# Study and Application of Urban Aquatic Ecosystem Health Evaluation Index System in River Network Plain Area

**DOI:** 10.3390/ijerph192416545

**Published:** 2022-12-09

**Authors:** Rui Ding, Kai Yu, Ziwu Fan, Jiaying Liu

**Affiliations:** 1Nanjing Hydraulic Research Institute, Nanjing 210029, China; 2Key Laboratory of Taihu Basin Water Resources Research and Management of Ministry of Water Resources, Nanjing 210029, China

**Keywords:** urban aquatic ecosystem health, evaluation index system, analytic hierarchy process, fuzzy comprehensive evaluation, river network plain area

## Abstract

The evaluation index system of urban aquatic ecosystem health is of great significance for the assessment and management of urban river networks, and for urban development planning. In this paper, the concept of urban aquatic ecosystem health was analyzed by the relationship between human, city and aquatic ecosystem, and its evaluation index system was established from environmental conditions, ecological construction, and social service. In addition, the weight value of each index was calculated by the analytic hierarchy process, and the grading standard of each index was set. Jiading New City, a typical city of the river network plain area in Yangtze River delta, was selected as the aquatic ecosystem health evaluation sample. The fuzzy comprehensive method was used to evaluate the aquatic ecosystem health of Jiading New City. The results indicated that the water ecosystem health of Jiading New City reached the “good” level. For the criterion level, environmental conditions and ecological construction reached the “good” level, and social services reached the “excellent” level. For the indicator level, most indicators reached “good” and “excellent” levels, but the river complexity and benthic macroinvertebrate diversity are still in the “poor” state, which indicates that the aquatic environment has greatly improved, but the aquatic ecosystem has not been fully restored. Results suggested that river complexity and biodiversity should be increased in urban construction planning. The evaluation index system established in this paper can be used to reflect the urban aquatic ecosystem health conditions in river network plain areas.

## 1. Introduction

Water is the source of life and the foundation of human survival and urban development [1]. In recent years, exploitation of water resource has been increasing with the rapid development of industrialization and urbanization [2]. Many pollutants were discharged into the urban river, resulting in many problems such as ecosystem degradation and environmental carrying capacity reduction [3]. The aquatic ecosystem in some urban areas has been seriously threatened [4]. How to improve the urban aquatic ecosystem has become one of the hot issues in international society [5,6]. Environmental evaluation is also crucial as the premise and tool of aquatic ecosystem protection and management.

The plain river network area owns a high-density river system and rich water resources [7], which is also generally one of the most densely populated and economically developed areas [8], resulting in heavy pollution loads. Water pollution and aquatic ecosystem degradation is more serious [9]. In the past few decades, with the fast development of urbanization, aquatic ecosystems in China have gone through stages of pollution and restoration [10]. With strong processes of pollution control and river regulation, the water quality of urban rivers has greatly improved, but aquatic ecosystem health is still not very good, and requires more restoration. The health evaluation index system of urban aquatic ecosystem in river network plain area is crucial for the ecosystem restoration.

There are many studies on the evaluation of urban ecosystems and aquatic ecosystems for rivers or lakes [11,12,13]. Zhang et al. [14] proposed a combined model for river health evaluation including physical, chemical, and biological elements; Chen et al. [15] evaluated riverine health using coordinated development degree model based on natural and social functions. Zhao et al. [16] constructed an urban river health evaluation index system based on PSR (Pressure-State-Response) framework. The coupling and coordination relationship between urban ecosystem and aquatic ecosystem has been established in previous studies. Xiong et al. [17] and Wang et al. [18] constructed a coupling coordination degree model between urban environments and ecosystems. However, very few studies have been conducted to establish the evaluation index system for aquatic ecosystem health on the urban scale. In addition, the aquatic ecosystem health index systems constructed in previous research focus on the aquatic ecosystem itself and its social service function for human beings. With the rapid process of the global urban ecological civilization, the construction of the urban aquatic ecosystem in the evaluation cannot be ignored.

Therefore, based on the previous research results and the conception of aquatic ecosystem health under the background of urbanization, the evaluation index system was established to objectively reflect the status of urban aquatic ecosystem and provide technical support for urban aquatic ecosystem restoration.

## 2. Materials and Methods

### 2.1. Study Region

Jiading New City is a typical river network plain area, with complex hydrological conditions, densely river network, and developed water system. There are 703 rivers, and the total length reaches 642.9 km. Jiading New City is one of the five key new cities in Shanghai, located in the northwest of Shanghai, with a total area of 161.7 km^2^ [19]. The river network in Jiading New City and arrangement of sampling points in field investigation are shown in Figure 1.

### 2.2. Index System Establishmen

#### 2.2.1. Conception of Urban Aquatic Ecosystem Health

Cities are closely related to the ecosystem. From the perspective of ecology, the city is an ecosystem with humans as the key element. From the perspective of urbanization, the ecosystem is the objective basis for human activities and urban development [20]. Water is the key to urban ecology, one of the most important resources of the urban ecosystem, the necessary condition for urban production, life, and ecology, and also the most critical environmental factor in the formation and development of cities [21]. The urban aquatic ecosystem provides services not only for cities, but also for humans, which is a key factor to connect cities and aquatic ecosystem. The relationship among city, human systems, and the biophysical aquatic ecosystem is very complicated, with mutual effects and constraints [22]. Cities are the main place for human activities, and their development depends on an aquatic ecosystem which provides social services, such as drinking water supplementation, and entertainment for humans. However, the neglect of urban water facility maintenance and unreasonable discharges of wastewater lead to water quality degradation, seriously affect the human habitat environment and restrict urban development [23]. The construction and maintenance of the urban aquatic ecosystem is quite essential to promote urban development. The relationship among human, cities, and the aquatic ecosystem is shown in Figure 2.

A healthy urban aquatic ecosystem includes a complete ecological structure and ecological construction to maintain a healthy water environment and social services to meet human needs, i.e., environmental condition, ecological construction, and social services in a “healthy” state. Healthy environmental condition refers to the satisfactory river network structure and shoreline structure [24], standard water quality, sufficient water quantity and self-purification ability [25], favorable biological integrity and biodiversity of water shoreline, and complete natural ecosystem structure. Healthy ecological construction is reflected in high-standard prevention planning, adequate control measures, and strict supervision of sewage discharge for the rapid development of urbanization. Health standards for social services mainly depend on the needs of human at different times. The current standards provide humans with high quality and adequate water supply services, water regulation services, water culture services, and other social services [26].

#### 2.2.2. Environmental Condition

The importance of the environmental condition of ecosystems is widely recognized [27,28]. The physical structure includes the pattern of the river and lake, aquatic ecological space, and ecological space of the shoreline [29,30]. Based on previous research on the structure of river and lake ecosystems, the urban aquatic ecosystem is characterized by water space, the aquatic environment, and the aquatic organism [31]. With regard to water space, the urban river network structure and connectivity, urban water surface area ratio [32], river connectivity, and river complexity [33] were selected as evaluation indexes. With regard to the aquatic environment, the guarantee rate of the ecological water level, sediment pollution, and self-purification ability [34] were selected to reflect the water quantity status, the water quality status at sediment, and the ability of the water to resist pollution, respectively. For aquatic organisms, benthic macroinvertebrate diversity, fish diversity, zooplankton diversity, and phytoplankton diversity were selected from the biological status indexes of different positions in urban rivers and lakes, such as the bottom, middle, and surface of the water body to reflect the aquatic ecosystem.

#### 2.2.3. Ecological Construction

Ecological construction is reflected in strengthening the protection of aquatic ecosystem and strict control of pollutant discharge [35]. The proportion of ecological revetment construction and the governance degree index of illegal use of river shorelines were selected for ecological protection. Urban water pollution mainly comes from the non-point source pollution caused by the use of urban agricultural fertilizers [36] and point source pollution caused by the daily life of residents, industrial development, and large-scale farming sewage discharge [37]. From the points of each pollution source, the annual chemical fertilizer use control, domestic sewage treatment rate, industrial wastewater discharge rate, and large-scale aquaculture sewage treatment rate were selected to reflect the control degree of water pollution discharges.

#### 2.2.4. Social Service

Health standards for social services depend on the needs of human activities, and are mainly reflected in flood control, resource supply, and river landscape culture [30]. Urban waterlogging in the river network plain area is quite frequent and causes huge property loss. Therefore, the function of urban flood control [38] and the influence of urban waterlogging [39] on social service function cannot be ignored. Flood and waterlogging control compliance rates were selected as indexes under the level of flood and waterlogging control. With regard to resource supply function, the water quality and water quantity of urban water supply were mainly considered [40], and the water quality compliance rate and water supply guarantee rate were selected, respectively. From the most intuitive point of view, human needs for water landscapes include water surface cleanliness and the connectivity of public riverbanks [41]; therefore, water surface cleanliness, water transparency, and riverbank connectivity were selected. In addition, public satisfaction [42], issued by questionnaire, should be considered as an evaluation index to reflect the residents’ overall satisfaction with the social service function of the urban aquatic ecosystem.

#### 2.2.5. Index System Establishment

Based on the above index analysis and following the principles of scientific, independence, operability, and quantification [43], the urban aquatic ecosystem index system was constructed, which included one target, three criteria, nine sub-criteria, and 25 indicators from aspects of environmental condition, ecological construction, and social service. The index system is shown in Table 1 below.

### 2.3. Evaluation Criteria

According to specifications, standards, norms, literature, and consultation with relevant experts [44,45,46], the values of evaluation standards were determined and divided into five grades: excellent, good, medium, poor, and very poor. The specific values of standards are shown in Table 2.

### 2.4. Evaluation Methods

The calculation method and data source of each index are shown in the Table 3.

The expert consultation method and analytic hierarchy process [47,48] were used to calculate the weight of each index. Firstly, the questionnaire was distributed to experts in various related fields. Then, the “1–9 scale method” was used to quantify the questionnaire results and construct the evaluation matrix. The maximum eigenvalue of the matrix and the corresponding eigenvector were calculated. Finally, the consistency test of the matrix was carried out. The eigenvector corresponding to the maximum eigenvalue of the matrix with satisfactory consistency was normalized as the weight vector.

The fuzzy comprehensive evaluation method [49] was used to evaluate the aquatic ecosystem of Jiading New City. It is a method to transform qualitative evaluation into quantitative evaluation based on fuzzy mathematical membership theory [50].Firstly, set two sets: the index set C = {C11, C12, …}, where Cij is the index j of sub-criterion Ci; the evaluation set P = {P1, P2, …}, where Pn is grade n of evaluation standards, and then the status value is substituted into the membership function through the fuzzy mapping of A-P to obtain the evaluation matrix of index membership degree. Membership degrees are calculated by the following equations:(1)Positive index (the larger the index value is, the healthier the aquatic ecosystem is)When *x*_i_ ≤ *S*_i1_, *r*_i1_ = 1, *r*_i2_ = *r*_i3_ = *r*_i4_ = *r*_i5_ = 0;When *S*_ij_ < *x*_i_ < *S*_ij+1_, *r*_ij_ = (*S*_ij+1_ − *x*_i_)/(*S*_ij+1_ − *S*_ij_), *r*_ij+1_ = (*x*_i_ − *S*_ij_)/(*S*_ij+1_ − *S*_ij_), the rest of *r* are zero;When *x*_i_ ≥ *S*_i5_, *r*_i5_ = 1, *r*_i1_ = *r*_i2_ = *r*_i3_ = *r*_i4_ = 0.(2)Negative index (the smaller the index value is, the healthier the aquatic ecosystem is)When *x*_i_ ≥ *S*_i1_, *r*_i1_ = 1, *r*_i2_ = *r*_i3_ = *r*_i4_ = *r*_i5_ = 0;When *S*_ij+1_ < *x*_i_ < *S*_ij_, *r*_ij_ = (*x*_i_ − *S*_ij+1_)/(*S*_ij_ − *S*_ij+1_), *r*_ij+1_ = (*S*_ij_ − *x*_i_)/(*S*_ij_ − *S*_ij+1_), the rest of *r* are zero;When *x*_i_ ≤ *S*_i5_, *r*_i5_ = 1, *r*_i1_ = *r*_i2_ = *r*_i3_ = *r*_i4_ = 0.where *x*_i_ is the value of index i; *S*_ij_ is *j*—level health standard of index *i*, *j* = 1, 2, …, 5, correspond to “very poor”, “poor”, “medium”, “good”, and “excellent” respectively; *r*_ij_ is relative membership grade of index *i* corresponding to grade *j*.

Combined with the weight value calculated above, the weighted average method was used to calculate membership degrees in other layers. The grade with the maximum membership degree was taken as the final fuzzy evaluation result, according to the principle of maximum membership degree.

## 3. Results

### 3.1. Indicator Calculation and Weight Results

All materials were mainly derived from existing data provided by local authorities and field investigation. The calculations, results, and data sources of each index in the index system are shown in Table 4.

Index weights calculated by the analytic hierarchy process are shown in Table 1.

### 3.2. Evaluation Results

The calculation results obtained by fuzzy comprehensive evaluation are shown in the Table 5 and Figure 3.

## 4. Discussion

According to the membership degree results, the aquatic ecosystem evaluation grade of Jiading New City is in the “good” state, and the membership degree of “excellent” (0.3638) is greater than “medium” (0.1434), indicating that it has the potential to improve from “good” to “excellent”. From the criterion level, environmental conditions and ecological construction are in the “good” state, and social services are in the “excellent” state.

For the sub-criterion layer of the environmental conditions, the evaluation of aquatic environment (C2) was significantly better than water space (C1) and aquatic organism (C3), which suggest that the aquatic environment, specifically water quality and sediment pollution, is recovering well, but the restoration of aquatic organisms still needs more time. It should be noted that benthic macroinvertebrate diversity (C34) is between medium and poor. The river complexity (C13) is quite essential for flow characteristics and biological habitat, but its evaluation is between medium and poor, which was caused by some historical developments. With the continuous development of urbanization, flood control and water resources development were usually a high priority for urban river networks, while the biological habitat of river networks was easily ignored. Urban river network regulation makes the river increasingly channelized, resulting in reduced heterogeneity in channel morphology and flow characteristics. Channelization constrains channel morphology, removes obstacles to flow, and shortens stream length. These modifications eliminate habitats in overflow areas such as wetlands and side channels. Once the river network is formed and constructed, it’s very hard to change the river complexity, which suggests that the concept of “water space”, specifically river complexity and water surface area ratio, should be a priority for urban construction planning.

For the sub-criterion layer of the ecological construction, the evaluation of water pollution control (C5) is significantly better than ecological protection (C4), which indicates that much attention was paid to water pollution control in the past decades, achieving good results. Ecological construction has just started and will be more and more important in the future.

For the sub-criterion layer of the ecological construction, the evaluation of flood and waterlogging control (C6) and water supply (C7) is better than water landscape (C8), which indicates that water safety and water supply is always the first priority, and that construction of water landscape still needs more attention.

There are several reasons for the evaluation results above, especially the river complexity and benthic macroinvertebrate diversity in the “poor” level. In the past decades, with the fast development of urbanization, aquatic ecosystems in China have gone through stages of pollution and restoration. At present, the water quality of urban rivers and lakes has greatly improved, but the aquatic ecosystem has not been fully restored. In addition, dredging is generally used for river regulation, but this physical measure will increase the destruction of benthic animal communities, resulting in a decline in diversity. It should be noted that benthic macroinvertebrates may be either clean species or tolerant species, because water chemistry can be an interesting way to increase the number of species and perhaps diversity in an environment enriched with some elements and with organic matter. More attention should be paid to the restoration of river and lake ecosystems. River complexity and biodiversity in urban construction planning should be increased to build a more healthy urban aquatic ecosystem.

To evaluate the reality and effectiveness of the model, Guidelines for river and lake health assessment by Ministry of Water Resources of China was used to evaluate the aquatic ecosystem evaluation grade of Jiading New City with regard to aquatic space, quantity, quality, organism and social service. The aquatic ecosystem evaluation grade of Jiading New City is also in the “good” state as a whole, which is consistent with the evaluation results of this paper. The evaluation model established in this paper is only for urban aquatic ecosystems in river network plain area. For other different environments in other river network plain areas, further study is still needed. At present, there are few studies on urban aquatic ecosystems. For the universality of the evaluation index system, further research and verification on the system’s applicability of multiple cities is needed.

## 5. Conclusions

According to the results of this study, the concept of urban aquatic ecosystem is defined by the core relationship between city, human systems, and the biophysical aquatic ecosystem. A healthy urban aquatic ecosystem includes the status of a aquatic ecosystem with a complete natural ecological system structure, the ecological construction to maintain aquatic ecosystem health, and the social service function to meet human needs.

Based on the health connotation of urban aquatic ecosystem, an index system of urban aquatic ecosystem, including one target, three criteria, nine sub-criteria and 25 indicators, was constructed. The aquatic ecosystem of Jiading New City was evaluated by a fuzzy comprehensive evaluation method.

The aquatic ecosystem of Jiading New City is in a “good” state and has the potential to improve from “good” to “excellent”. The environmental condition, ecological construction, and social service have reached the “good” or “excellent” level. However, some indicators, such as river complexity and benthic macroinvertebrate diversity, are still in the “poor” state. More attention should be paid to the restoration of river and lake ecosystems in river network plain areas. Results suggested that increased attention to river complexity and biodiversity in urban construction planning will help to build a more healthy urban aquatic ecosystem.

## Figures and Tables

**Figure 1 ijerph-19-16545-f001:**
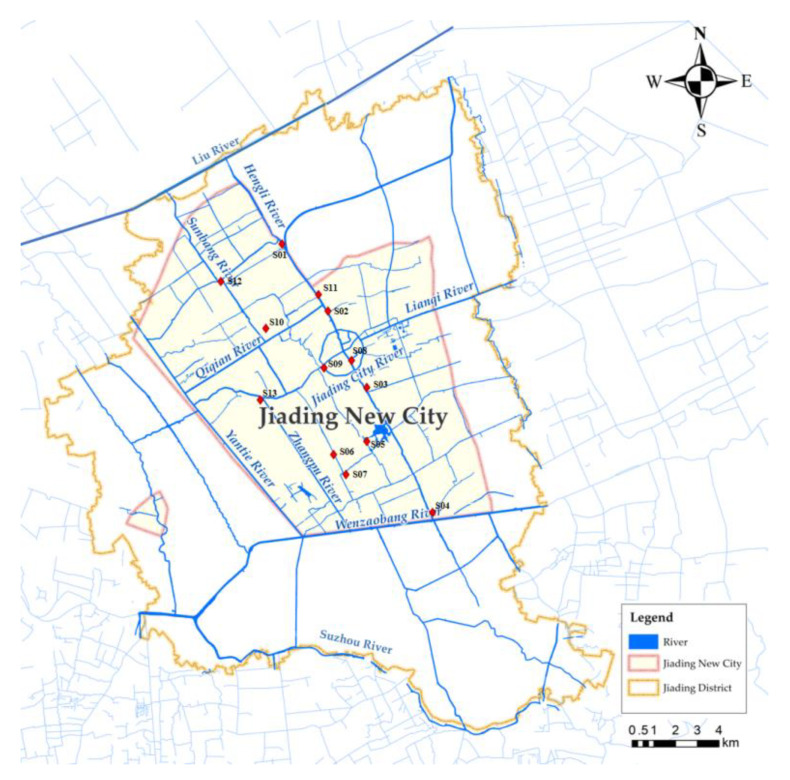
Water system diagram of Jiading New City.

**Figure 2 ijerph-19-16545-f002:**
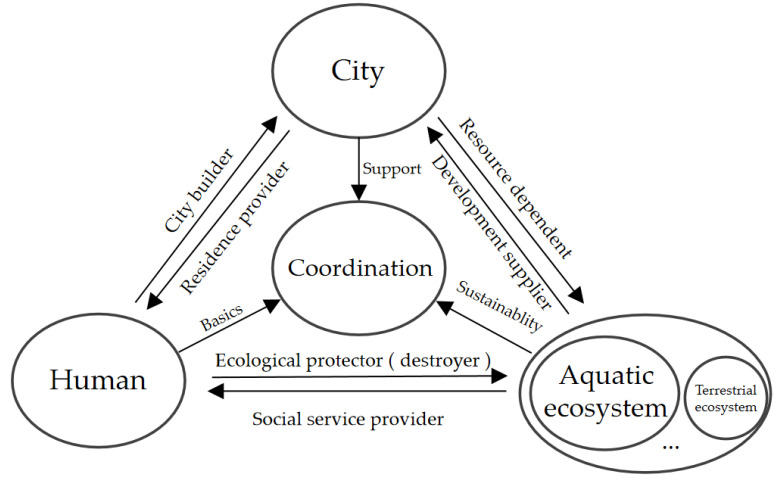
Relationship among human, city, and the aquatic ecosystem.

**Figure 3 ijerph-19-16545-f003:**
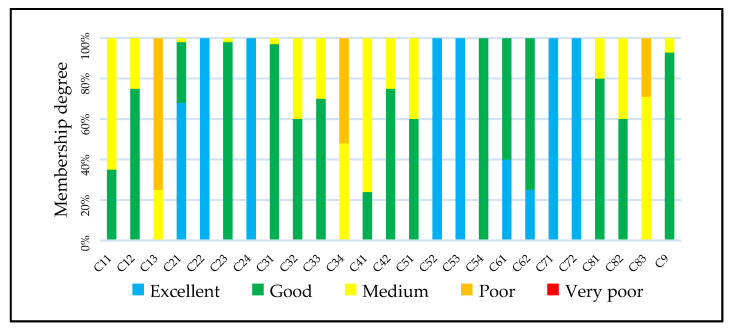
Index evaluation grade membership degree.

**Table 1 ijerph-19-16545-t001:** Index system, weights of indexes, and ranking.

Target Layer	Criterion Layer (Weights)	Sub-Criterion Layer (Weights)	Index Layer	Index Weights			
				The Weights Relative to the Sub-Criteria Layer	The Weights Relative to the Criterion Layer	The Weights Relative to the Target Layer	Ranking
A Urban aquatic ecosystem	B1 Environmental condition (0.3865)	C1 Water space (0.2626)	C11 Urban water surface area ratio	0.2736	0.0719	0.0278	16
			C12 River connectivity	0.4762	0.1251	0.0483	6
			C13 River complexity	0.2502	0.0657	0.0254	19
		C2 Aquatic environment (0.3258)	C21 Water quality	0.3308	0.1078	0.0417	7
			C22 Sediment pollution	0.2144	0.0699	0.0270	17
			C23 Self-purification ability	0.3020	0.0984	0.0380	10
			C24 Guarantee rate of ecological water level	0.1527	0.0498	0.0192	21
		C3 Aquatic organism (0.4115)	C31 Phytoplankton diversity	0.3048	0.1254	0.0485	5
			C32 Zooplankton diversity	0.2556	0.1052	0.0407	9
			C33 Fish diversity	0.2057	0.0847	0.0327	14
			C34 Benthic macroinvertebrate diversity	0.2338	0.0962	0.0372	11
	B2 Ecological construction (0.1726)	C4 Ecological protection (0.4345)	C41 Proportion of ecological revetment construction	0.4494	0.1953	0.0337	13
			C42 Governance of illegal use of water shoreline	0.5506	0.2392	0.0413	8
		C5 Water pollution control (0.5655)	C51 Annual chemical fertilizer use control	0.1857	0.1050	0.0181	22
			C52 Domestic sewage treatment rate	0.2677	0.1514	0.0261	18
			C53 Industrial wastewater discharge rate	0.2982	0.1686	0.0291	15
			C54 Large-scale aquaculture sewage treatment rate	0.2484	0.1404	0.0242	20
	B3 Social service (0.4410)	C6 Flood and waterlogging control (0.4624)	C61 Flood control compliance rate	0.6250	0.2890	0.1274	1
			C62 Waterlogging control compliance rate	0.3750	0.1734	0.0765	4
		C7 Water supply (0.3718)	C71 Water quality compliance rate	0.5327	0.1981	0.0873	2
			C72 Water supply guarantee rate	0.4673	0.1737	0.0766	3
		C8 Water landscape (0.0870)	C81 Water surface cleanliness	0.3929	0.0342	0.0151	24
			C82 Water sensory transparency	0.4480	0.0390	0.0172	23
			C83 Riverbank connectivity	0.1591	0.0138	0.0061	25
		C9 Public satisfaction (0.0789)	C9 Public satisfaction	1.0000	0.0789	0.0348	12

**Table 2 ijerph-19-16545-t002:** Evaluation criteria.

Indexes (Units)	Excellent	Good	Medium	Poor	Very Poor
C11 (%)	12	10	8	6	4
C12 (%)	100	80	60	40	0
C13	1.14	1.12	1.08	1.04	1.02
C21	I–III	IV	V	Inferior V	Black-odorous water
C22	1	1.5	2	2.5	3
C23 (mg/L)	7.5	6	5	3.5	0
C24 (%)	95	90	85	80	75
C31	3	2	1	0.5	0
C32	3	2	1	0.5	0
C33	3	2	1	0.5	0
C34	3	2	1	0.5	0
C41 (%)	100	80	60	40	0
C42 (%)	100	80	60	40	0
C51 (%)	10	5	0	−5	−10
C52 (%)	95	90	85	80	75
C53 (%)	100	95	90	85	80
C54 (%)	95	90	85	80	75
C61 (%)	95	90	85	80	50
C62 (%)	100	80	60	50	40
C71 (%)	95	80	65	50	40
C72 (%)	95	85	60	40	20
C81 (m^2^)	0.5	1	2	3	4
C82 (cm)	80	60	40	30	20
C83 (%)	90	80	60	50	40
C9 (grade)	95	85	60	40	0

**Table 3 ijerph-19-16545-t003:** Calculation method and data sources of each index.

Indexes (Units)	Calculations	Data Sources
C11 (%)	Surface area under normal water level/study area × 100% (by Remote sensing)	Water conservancy bureau
C12 (%)	Average annual opening time percentage of sluice	
C13	River length/length of a straight line from the beginning to the end of a river (by Remote sensing)	
C21	Evaluation of dissolved oxygen, ammonia nitrogen, biochemical oxygen demand, permanganate index, total phosphorus, total nitrogen (Environmental Quality Standards for Surface Water GB 3838-2002 and Technical Specifications for Automatic Monitoring of Surface Water HJ 915-2017)	Hydrographic office
C22	Concentration of maximum pollutants in sediment/corresponding standard value	Field investigation
C23 (mg/L)	Dissolved oxygen concentration	Hydrographic office
C24 (%)	Proportion of control section samples with ten-day average water level greater than or equal to guarantee target	Water conservancy bureau
C31	Shannon-Wiener biodiversity index [26]	Field investigation
C32		
C33		
C34		
C41 (%)	Length of ecological shoreline for river reconstruction/total length of river shoreline × 100%	Water conservancy bureau
C42 (%)	The number of shorelines used in regulated illegal waters/number of illegal use × 100%	
C51 (%)	(Fertilizer use in previous period/fertilizer use in the current period−1) × 100%	Water conservancy
C52 (%)	Domestic sewage treatment quantity/total amount of sewage × 100%	
C53 (%)	Standard industrial wastewater discharge/total amount of sewage × 100%	
C54 (%)	Sewage treatment amount of large-scale aquaculture/total amount of sewage × 100%	
C61 (%)	The embankment length reached the flood control project/total length of embankment × 100%	Hydrographic office
C62 (%)	Weighted average of each polder area meeting flood control standard	
C71 (%)	Number of qualified drinking water sources/total drinking water sources × 100%	Hydrographic office
C72 (%)	Number of days reaching guaranteed water level or flow/365 × 100%	
C81 (m^2^)	Cumulative area of surface garbage per 5000 m^2^ water area	Field investigation
C82 (cm)	Measure transparency of observation section by secchi disc	
C83 (%)	River-lake shoreline penetration length/total length of river and lake shoreline × 100%	Hydrographic office
C9 (grade)	The proportion of the people whose scores on the questionnaire including water scape, flood control, drinking water and other indicators that reflect people’s needs are more than the “satisfactory” level	Questionnaire

**Table 4 ijerph-19-16545-t004:** Calculation results.

Indexes (Units)	Results	Indexes (Units)	Results
C11 (%)	8.7	C51 (%)	3
C12 (%)	75	C52 (%)	95
C13	1.05	C53 (%)	100
C21	III 67.82% IV 29.88% V 2.30%	C54 (%)	90
C22	0.9	C61 (%)	92
C23 (mg/L)	5.98	C62 (%)	85
C24 (%)	95	C71 (%)	100
C31	1.97 (Major creatures: *Skeletonema sp*. and *Merismopedia convolute*)	C72 (%)	95
C32	1.60 (Major creatures: *Diaphanosoma brachyurum*)	C81 (m^2^)	1.2
C33	1.70	C82 (cm)	52
C34	0.74 (Major creatures: *Limnodrilus hoffmeisteri* and *Bellamya aeruginosa*)	C83 (%)	57.1
C41 (%)	64.8	C9 (grade)	83.2
C42 (%)	75		

**Table 5 ijerph-19-16545-t005:** Membership degree distribution and evaluation grade.

Factors			Excellent	Good	Medium	Poor	Very Poor	Evaluation Grade
A	B1	C1	0	0.4529	0.3594	0.1877	0	Good
		C2	0.5921	0.3952	0.0127	0	0	Excellent
		C3	0	0.5930	0.2854	0.1216	0	Good
		Overall situation	0.1929	0.4918	0.2160	0.0993	0	Good
	B2	C4	0	0.5208	0.4792	0	0	Good
		C5	0.5659	0.3598	0.0743	0	0	Excellent
		Overall situation	0.3200	0.4298	0.2502	0	0	Good
	B3	C6	0.3438	0.6562	0	0	0	Good
		C7	1	0	0	0	0	Excellent
		C8	0	0.5832	0.3707	0.0461	0	Good
		C9	0	0.9280	0.0720	0	0	Good
		Overall situation	0.5307	0.4273	0.0380	0.0040	0	Excellent
	Overall situation		0.3638	0.4527	0.1434	0.0402	0	Good

## Data Availability

Not applicable.

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
