# Peer review of "Study and Application of Urban Aquatic Ecosystem Health Evaluation Index System in River Network Plain Area"

_ijerph, 2022, doi:10.3390/ijerph192416545_

Round 1

Reviewer 1 Report (Previous Reviewer 2)

The authors have satisfactorily changed the text of the manuscript

This manuscript is a resubmission of an earlier submission. The following is a list of the peer review reports and author responses from that submission.

Round 1

Reviewer 1 Report

This paper is of considerable interest to researchers of urban aquatic ecosystem health, especially those outside China, where it appears that almost all of the research on the evaluation index system appears to have been undertaken to date. 

Please note that I do not have technical competency in many of the methods used in the study.  I have a general ecology background and interest in urban aquatic ecosystem health as part of planning for urban sustainability, hence my interest to review this paper.  I have a largely positive impression of the paper.  The results are plausible even without technical expertise in the methods of their derivation. They are significant in their application to urban sustainability research and decision-making, because of the comprehensiveness of the list of sub-criteria and indices shown in Table 1, especially reflecting biophysical, ecological and social factors, and investigation through both quantitative and qualitative methods.

There are elements of circularity in some of the arguments in the paper.  For example, between the urban aquatic ecosystem and other elements of the urban system, as commented below regarding Fig. 2.  Also between some of the urban aquatic ecosystem indices and ecological construction and social service indices. This means that it is not possible to regard the indices in Table 2 as being independent.  i.e. C4 and C5 indices are to varying extent dependent on C1-C3 indices; similarly with C6-C9 indices and C1-C5 indices.  I would expect to see some discussion of the extent to which the fuzzy logic methodology depends on or deals with dependent and independent variables.

Because of my interests and expertise my comments are principally aimed at the paper's context, introduction and the presentation of the methodology.  I would also recommend that the authors pay attention to English language  and grammar aspects, seeking assistance for this aspect if necessary,  in order to increase the accessibility of the paper to the wider international readership.

Specific  comments:

Line 3: "plain river" should be "river plain" throughout

Line 46: Some of the references for the the health evaluation index system should be introduced at this point rather than later.

Lines 63-64: the EIS is established only for the study area in this paper, not generally.

Lines 78-79: humans are not the main body"(element)" of the city system from an ecological perspective. The foundation comes from the terrestrial and aquatic environment even if humans dominate that environment.

Fig 2: a useful conceptualisation, but omits other elements of the urban envt such as land, soils, other biota than humans.  The environmental envelope concept stemming back to Herman Daly and going right through to the planetary boundaries work of the Stockholm Resilience Centre is a more appropriate way of illustrating the conceptualisation attempted in Fig 1. 

L 113-122: I would expect riparian ecosystem indicators (composition, functioning etc, other than just connectivity-related) to be included in the otherwise impressive list of factors in Table 1.

L 139: A little more on what is meant by "Flood and waterlogging control compliance rate" and the difference between them would be useful given the importance of these two factors.  At least some references should be given.

L 150-154: a reference forward to section 2.4 should be given as that section contains a lot of important information on the contents of Table 1.

Section 2.4 especially lines 170-174: as mentioned earlier I am not an expert in the methodology but I can see its relevance to the significance of the paper.  Therefore a couple of suggestions to making the paper more accessible to non-experts would be a) to be more explicit about which data sets presented in the paper constitute the factor set and the evaluation set used for the analysis; and b) to include a definition or general comment about the nature of fuzzy set analysis and membership degree analysis (or at least some introductory references).

Table 3, last line, re C9: How does this result  (grade of 83.2%) reflect the actual questions and answers in the questionnaire and the description in lines 147-148?

L 204-26: Clarify how the membership degree of “excellent” being greater than “medium” indicates that the parameter has the potential to improve from “good” to “excellent”

L208-224: A comment on the effect of channelisation on connectivity in the river network would be helpful.

L231: I think that "ecological construction" should read "social services"

L 247-248:  As written the sentence is circular. A phrase such as "the concept of the urban aquatic ecosystem is defined by the core relationships between city and human systems, and the biophysical aquatic ecosystem" would better express the concepts attempted to be illustrated in Fig 2, and my previous comment on that figure. 

L256-257: This sentence should be qualified by the phrase "according to the results of this study,..."

Author Response

Response to Reviewer 1 Comments

Point 1: Line 3: "plain river" should be "river plain" throughout 

Response 1: We have modified in this article, see the article for details.

Point 2: Line 46: Some of the references for the the health evaluation index system should be introduced at this point rather than later.

Response 2: We have made corresponding adjustments and add relevant references.

References:

11.Wan,X.;Yang,T.;Zhang,Q.;Yan,X.;Zheng,Y.A novel comprehensive model of set pair analysis with extenics for river health evaluation and prediction of semi-arid basin - A case study of Wei River Basin, China. Science of The Total Environment, 2021,775(8):145845.

12.Li,T.;Huang,X.;Jiang,X.;Wang,X.Assessment of ecosystem health of the Yellow River with fish index of biotic integrity.Hydrobiologia, 2018, 814(1):31-43.

13.Qiting,Z.;Hao,C.;Yongyong,Z.Impact factors and health assessment of aquatic ecosystem in Upper and Middle Huai River Basin. journal of hydraulic engineering, 2015, 46(9):1019-1027.

Point 3: Lines 63-64: the EIS is established only for the study area in this paper, not generally.

Response 3: Our article mainly aims at establishing the index system based on the common characteristics of urban aquatic ecosystem in river network plain area, and then applied it to Jiading New City.

Point 4: Lines 78-79: humans are not the main body"(element)" of the city system from an ecological perspective. The foundation comes from the terrestrial and aquatic environment even if humans dominate that environment.

Response 4: We have modified “the main body” to “the key element” in this article.

Point 5: Fig 2: a useful conceptualisation, but omits other elements of the urban envt such as land, soils, other biota than humans.  The environmental envelope concept stemming back to Herman Daly and going right through to the planetary boundaries work of the Stockholm Resilience Centre is a more appropriate way of illustrating the conceptualisation attempted in Fig 1.

Response 5: We have added terrestrial ecosystem in addition to aquatic ecosystem in Fig 1.See the article for details.

Point 6: L 113-122: I would expect riparian ecosystem indicators (composition, functioning etc, other than just connectivity-related) to be included in the otherwise impressive list of factors in Table 1.

Response 6: We have considered the riparian ecosystem in these indexes including C41 proportion of ecological revetment construction and C83 riverbank connectivity.

Point 7: L 139: A little more on what is meant by "Flood and waterlogging control compliance rate" and the difference between them would be useful given the importance of these two factors.  At least some references should be given.

Response 7: We have added some references accordingly.

References:

38.Fu,G.A fuzzy optimization method for multicriteria decision making: An application to reservoir flood control operation. Expert Systems with Applications,2008,34(1):145-149.

39.Zhang,Q.;Wu,Z.;Guo,G.;Zhang,H.;Tarolli,P.Explicit the urban waterlogging spatial variation and its driving factors: The stepwise cluster analysis model and hierarchical partitioning analysis approach.Science of The Total Environment,2020, 763(July 2017):143041.

Point 8: L 150-154: a reference forward to section 2.4 should be given as that section contains a lot of important information on the contents of Table 1.

Response 8: We did have few references in this section, and we have added relevant references.

References:

40.Qian,J.Z.;Ru,L.;Wang,J.Q.;Yu,X.L.Environmental health risk assessment for urban water supply source.Journal of Hydraulic Engineering,2004:90-93.(In Chinese)

41.Mcmahon,J.M.;Olley,J.M.;Brooks,A.P.;Smart,J.C.R.;Stout,J.C.Vegetation and longitudinal coarse sediment connectivity affect the ability of ecosystem restoration to reduce riverbank erosion and turbidity in drinking water.Science of The Total Environment,2020,707:135904.

42.Peng,B.;Wang,H.;Wang,R.;Han,Y.;Huang,W.;Zhu,Y.Development of a River Health Assessment System for the Lower Yellow River.Journal of Hydroecology,2014,35(6):81-87.

Point 9: Section 2.4 especially lines 170-174: as mentioned earlier I am not an expert in the methodology but I can see its relevance to the significance of the paper.  Therefore a couple of suggestions to making the paper more accessible to non-experts would be a) to be more explicit about which data sets presented in the paper constitute the factor set and the evaluation set used for the analysis; and b) to include a definition or general comment about the nature of fuzzy set analysis and membership degree analysis (or at least some introductory references).

Response 9: We have made some supplementary explanations of data sets and added a reference.

Reference:

Lai,C.G.;Chen,X.H.;Chen,X.Y.;Wang,Z.L.;Wu,X.S.;Zhao,S.W.A fuzzy comprehensive evaluation model for flood risk based on the combination weight of game theory.Nat.Hazards,2015, 77, 1243–1259

Point 10: Table 3, last line, re C9: How does this result (grade of 83.2%) reflect the actual questions and answers in the questionnaire and the description in lines 147-148?

Response 10: In this article, the public satisfaction is defined as the proportion of the people whose scores on the questionnaire are more than the “satisfactory” level. The questionnaires includes water scape, flood control, drinking water quality, and other indicators that reflect people's needs.What we explained in Table 3 is not particularly clear, so we have given more explanations in it.

Point 11: L 204-26: Clarify how the membership degree of “excellent” being greater than “medium” indicates that the parameter has the potential to improve from “good” to “excellent”

Response 11: According to the formula for calculating the membership degree, the membership degree can be equal to the proximity between the index value and different evaluation grades, so the membership degree of “excellent” being greater than “medium” numerically indicates that the parameter has the potential to improve from “good” to “excellent”

Point 12: L 208-224: A comment on the effect of channelisation on connectivity in the river network would be helpful.

Response 12: In this article, river connectivity is reflected by the opening time percentage of sluice .The channelization mainly affects the complexity of rivers.

Point 13: L231: I think that "ecological construction" should read "social services"

Response 13: Ecological construction more reflects the protection and construction of aquatic ecosystem given by human, and social services more reflects the help and feedback given by aquatic ecosystem to human. So we distinguished them in this article.

Point 14: L 247-248: As written the sentence is circular. A phrase such as "the concept of the urban aquatic ecosystem is defined by the core relationships between city and human systems, and the biophysical aquatic ecosystem" would better express the concepts attempted to be illustrated in Fig 2, and my previous comment on that figure.

Response 14: Thank you for your valuable advice, your description can express the concepts better and more clearly. We have made corresponding modification in the article.See the article for details.

Point 15: L256-257: This sentence should be qualified by the phrase "according to the results of this study,..."

Response 15: Thank you for pointing out our informal English usage, we have modified it in the article.

Reviewer 2 Report

1 - In fact, the authors bring a very interesting proposal, but that deviates a little
from the traditional indices used to predict the quality of aquatic ecosystems.

The proposal presented intends to gather numerous variables from several sectors in a single index, which seems to be a very difficult task.
In my opinion to assess the health of the aquatic ecosystem, the main layers are C2 and C3,
with indices from C21 to C34. Which already makes the entire evaluation process extremely complex.

2 – In Material and Methods, the manuscript does not show how the collections
and analyzes of each of the items evaluated were performed.
This methodological description is fundamental.

3 - Only the C21 (Water Quality) index, in large countries, is already very difficult
to interpret, especially for a country like China that must have waters with different
chemical and biological characteristics in its different regions.
Here we have a big question that is the question of “quality”. Which would have the best quality?
The one that is natural, without contamination or release, or the one that receives several releases,
but which has variables within the maximum allowed values? And from that point of view, who is the water for?
For the community that lives there, or for human use?

4 - Under these questions, for the C21 index only 4 variables were determined (Table 3) which is practically nothing
in terms of chemical evaluation of a water body.

Would this be enough to guarantee a positive or negative assessment in terms of Water Quality?

5 - Regarding the C31 to C34 indices, it is observed that C31 and C32 have the highest weights
in the evaluation and may be the result of a eutrophic environment, where a large load
of organic matter and sewage is being released.

How to consider that this would be positive in terms of ecosystem health?

6 - In the discussion (L. 208-213) a question is raised related to the low diversity of macroinvertebrates.
The explanation would be that macroinvertebrates are resistant to pollution.
It should be noted that the diversity of species is related to the characteristics of the environment,
sediment or water. Often this diversity in clean environments can be lower than in polluted environments,
since in an environment enriched with some elements and with organic matter, water chemistry can be interesting
to increase the number of species and perhaps diversity. I think this observation point is missing from the discussion.

7 - The conclusion of the manuscript leaves something to be explained in terms of the evaluation of the index presented.
It is understood that a proposal was presented and implemented, but the reality and effectiveness of the idea presented
were not discussed or determined. How to evaluate the effectiveness of a model without comparing it in different environments?
How to know if the model works, without assessing the condition of an aquatic ecosystem in “excellent health”?

Author Response

Response to Reviewer 2 Comments

Point 1: In fact, the authors bring a very interesting proposal, but that deviates a little from the traditional indices used to predict the quality of aquatic ecosystems. The proposal presented intends to gather numerous variables from several sectors in a single index, which seems to be a very difficult task. In my opinion to assess the health of the aquatic ecosystem, the main layers are C2 and C3, with indices from C21 to C34. Which already makes the entire evaluation process extremely complex. 

Response 1: For the natural aquatic ecosystem, I think these indexes (C21 to C34) are quite suitable, but for the urban aquatic ecosystem which includes human civilization, we think indexes reflecting the relationship between human beings and aquatic ecosystem still need to be added.

Point 2: In Material and Methods, the manuscript does not show how the collections and analyzes of each of the items evaluated were performed. This methodological description is fundamental.

Response 2: In Table 3, we introduced the acquisition and calculation methods of indicators, and the calculation methods of some indicators have been expanded.

Point 3: Only the C21 (Water Quality) index, in large countries, is already very difficult to interpret, especially for a country like China that must have waters with different chemical and biological characteristics in its different regions. Here we have a big question that is the question of “quality”. Which would have the best quality? The one that is natural, without contamination or release, or the one that receives several releases, but which has variables within the maximum allowed values? And from that point of view, who is the water for? For the community that lives there, or for human use?

Response 3: We referenced the environmental quality standards for surface water GB 3838-2002, and made minor adjustments to the evaluation grade for urban aquatic ecosystem.

Point 4: Under these questions, for the C21 index only 4 variables were determined (Table 3) which is practically nothing in terms of chemical evaluation of a water body. Would this be enough to guarantee a positive or negative assessment in terms of Water Quality?

Response 4: According to the environmental quality standards for surface water GB 3838-2002 and technical specifications for automatic monitoring of surface water HJ 915-2017, there are 9 main indexes including water temperature, pH, turbidity, conductivity, dissolved oxygen, ammonia nitrogen, permanganate index, total phosphorus, total nitrogen for evaluating water quality, of which the most important and most commonly used are chemical oxygen demand, permanganate index, ammonia nitrogen, and total phosphorus.

Point 5: Regarding the C31 to C34 indices, it is observed that C31 and C32 have the highest weights in the evaluation and may be the result of a eutrophic environment, where a large load of organic matter and sewage is being released. How to consider that this would be positive in terms of ecosystem health?

Response 5: For aquatic vegetation, they can perform photosynthesis and provide dissolved oxygen. So they play an active role in aquatic ecosystem health.For phytoplankton density, a moderate amount of phytoplankton can prevent algae growth, does not cause eutrophication, and also plays a positive role in the composition of the entire food web.

Point 6: In the discussion (L. 208-213) a question is raised related to the low diversity of macroinvertebrates. The explanation would be that macroinvertebrates are resistant to pollution. It should be noted that the diversity of species is related to the characteristics of the environment, sediment or water. Often this diversity in clean environments can be lower than in polluted environments, since in an environment enriched with some elements and with organic matter, water chemistry can be interesting to increase the number of species and perhaps diversity. I think this observation point is missing from the discussion.

Response 6: According to the results of field investigation to macroinvertebrates, there mainly exists Limnodrilus hoffmeisteri and Bellamya aeruginosa in the study area, and Limnodrilus hoffmeisteri is pollution-tolerant species. So we come to this conclusion.

Point 7: The conclusion of the manuscript leaves something to be explained in terms of the evaluation of the index presented. It is understood that a proposal was presented and implemented, but the reality and effectiveness of the idea presented were not discussed or determined. How to evaluate the effectiveness of a model without comparing it in different environments? How to know if the model works, without assessing the condition of an aquatic ecosystem in “excellent health”?

Response 7: We held a meeting with the local environmental protection bureau and water affairs bureau in Jiading to discuss the situation, showing that the results could better reflect the real situation. At present, there are few researches on urban aquatic ecosystem. For the universality of the evaluation index system, it also requires researchers further verify the applicability of multiple cities.

Round 2

Reviewer 2 Report

Almost all questions were not satisfactorily answered.

In the answer to question 1 the authors simply confirm my question and inform that new data need to be added to the work.

Question 2 was not aswered, as the answer included was for a different aspect than the one requested in the question.

Sorry, but in question 3 the authors were not asked to change the text or to cite their references, but rather to discuss the points observed in the question.

In the answer to question 4, the authors clearly show the reference on which they based themselves to build their index. Unfortunately, the question remains because with the 4 variables mentioned it is not possible to talk about water quality.

In the case of question 5 there is a serious problem of understanding the excessive growth of algae or plants in an aquatic environment does not represent a good healthy environment, but and environment contaminated with nutrients that cause eutrophication of the waters. Then, the question remains.

Regarding the response to question 6, who said that Limnodrilus hoffmeisteri and Bellamya aeruginosa are pollution tolerant? What kind of pollution? Where's the citation in text? This is a discussion or a statement.

The question 7 remains. How to evaluate the efectiveness of a model without making comparisons between a contaminated environment and a clean environment? How to know if the model is good, without comparing it to a "real" aquatic ecosystem with "excellent health"?